# INFERENCE-COST-AWARE DYNAMIC TREE CONSTRUCTION FOR EFFICIENT INFERENCE IN LARGE LANGUAGE MODELS

**Yinrong Hong**[1]\*, **Zhiquan Tan**[2]\*, **Kai Hu**[1]†
1: Beihang University
2: E Fund Management Co., Ltd.

## ABSTRACT

Large Language Models (LLMs) face significant inference latency challenges stemming from their autoregressive design and large size. To address this, speculative decoding emerges as a solution, enabling the simultaneous generation and validation of multiple tokens. While recent approaches like EAGLE-2 and EAGLE-3 improve speculative decoding using dynamic tree structures, they often neglect the impact of crucial system variables such as GPU devices and batch sizes.

Therefore, we introduce a new dynamic tree decoding approach called CAST that takes into account inference costs, including factors such as GPU configurations and batch sizes, to dynamically refine the tree structure. Through comprehensive experimentation across six diverse tasks and utilizing six distinct LLMs, our methodology demonstrates remarkable results, achieving speeds up to 5.2 times faster than conventional decoding methods. Moreover, it generally outperforms existing state-of-the-art techniques from 5% to 20%. The code is available at https://github.com/EAGLE-Research/sglang-eagle4.

## 1 INTRODUCTION

Large Language Models (LLMs) have showcased remarkable capabilities (OpenAI, 2023; Touvron et al., 2023) and are extensively applied across various domains. Nevertheless, the vast scale of their parameters, often exceeding hundreds of billions, presents notable challenges. This is particularly evident during the autoregressive text generation process, where generating each token involves referencing previous tokens, resulting in considerable latency. In real applications like chatbots, producing hundreds to thousands of tokens can render LLM inference slow and resource-intensive. This shows the need for better inference speedup methods.

To address this challenge, speculative decoding techniques (Leviathan et al., 2023; Chen et al., 2023) have emerged. These methods swiftly generate initial tokens and validate them concurrently, thereby diminishing inference latency by generating multiple tokens in a single forward pass. While traditional speculative decoding adopts a chain-structured draft, recent progressions have introduced tree-structured drafts to boost efficiency. For example, EAGLE (Li et al., 2024b) utilizes a static draft tree structure, incorporating a fixed number of candidates at each stage. However, this fixed approach overlooks the context-specific nature of token acceptance rates, contradicting the fundamental premise of speculative sampling that simpler tokens can be predicted by smaller models. Subsequently, EAGLE-2 (Li et al., 2024c) and EAGLE-3 (Li et al., 2025) leverage dynamic trees to further enhance performance.

While EAGLE-2 and EAGLE-3 have begun to harness the potential of dynamic tree structures, they fall short in adapting the structure based on crucial factors like GPU devices and batch sizes. Our proposed unified approach tackles this limitation by modeling the effects of variables such as device type and batch sizes as costs. The motivation behind our approach is that a higher number of tokens does not always equate to better performance. Taking into account the inference cost, there exists

---

[1]These authors contributed equally.
[2]Corresponding author.

a critical value beyond which adding more tokens becomes inefficient, slowing down the overall process. Drawing on these insights, we introduce a novel cost-conscious strategy that dynamically determines the tree's depth, token count per layer, and the number of tokens to be validated by the target model. Integrating this inference cost-aware dynamic tree construction method with the cutting-edge technique EAGLE-2 or EAGLE-3 yields an advanced method: Cost-Aware Speculative Tree (CAST). This method adjusts the draft tree structure dynamically by balancing the trade-off between accepted token numbers and inference cost, resulting in accelerated speedups.

Our comprehensive evaluations span six distinct tasks: multi-turn conversation, code generation, mathematical reasoning, instruction following, summarization, and question answering. The datasets utilized encompass MT-bench (Zheng et al., 2023), HumanEval (Chen et al., 2021), GSM8K (Cobbe et al., 2021), Alpaca (Taori et al., 2023), CNN/Daily Mail (Nallapati et al., 2016), and Natural Questions (Kwiatkowski et al., 2019). We benchmark our method against state-of-the-art speculative decoding techniques: standard speculative decoding (Joao Gante, 2023; Leviathan et al., 2023; Chen et al., 2023), Medusa (Cai et al., 2024), PLD (Saxena, 2023), Lookahead (Fu et al., 2023), EAGLE (Li et al., 2024b), EAGLE-2 (Li et al., 2024c), and EAGLE-3 (Li et al., 2025). Experiments are conducted across various LLM series with different batch sizes, including Vicuna, LLaMA3, Qwen2, and distilled DeepSeek-R1. Our method consistently surpasses all baseline approaches, achieving speedups of up to 5.2x and typically delivering speed enhancements ranging from $5\%$ to $20\%$ compared to the previous state-of-the-art method.

In summary, our paper offers the following contributions:

- We propose a new dynamic-tree-based speculative decoding method CAST based on the trade-off between the number of tokens to be verified and the inference cost.

- The proposed method generalizes previous state-of-the-art methods EAGLE-2 and EAGLE-3 and also systematically considers the impact of batching and GPU, which is less discussed in the literature.

- We conduct extensive experiments among 6 tasks and 6 models. The proposed method usually achieves $5 - 20\%$ speedup than the previous SOTA method and up to *5.2x* speedup than the vanilla autoregressive method.

## 2 RELATED WORKS

**Speculative Decoding**   The goal of speculative decoding is to accelerate LLM inference without losing output quality. Its core idea is to separate proposal from verification: a lightweight draft model suggests tokens, and the base LLM validates them. This shifts much of the workload to the draft model while preserving consistency, reducing latency compared with conventional step-by-step decoding.

Early work focused on greedy decoding. Stern et al. (2018) introduced blockwise decoding and Sun et al. (2021) proposed instantaneous methods, both allowing multiple tokens per step. Later, speculative sampling (Leviathan et al., 2023; Chen et al., 2023) extended the idea to non-greedy settings, establishing its broad applicability.

Subsequent methods improved draft efficiency and base-model alignment. SpecInfer (Miao et al., 2023) used draft-model ensembles and tree-mask attention. Medusa (Cai et al., 2024) leveraged MLPs on internal states to predict multiple tokens. EAGLE (Li et al., 2024b) expanded tree proposals for higher acceptance. Draft-and-Verify frameworks (Zhang et al., 2023; Hooper et al., 2023; Yang et al., 2023; Monea et al., 2023; Li et al., 2024a; Yi et al., 2024; Liu et al., 2024; Sun et al., 2024; Elhoushi et al., 2024; Svirschevski et al., 2024) introduced early exits and partial model reuse, partitioning the LLM into fast generators and verifiers.

More recently, dynamic draft trees emerged. GLIDE and CAPE (Du et al., 2024) added fallback branches for uncertain cases but limited expansion. EAGLE-2 (Li et al., 2024c) removed such constraints for fully adaptive growth, while EAGLE-3 (Li et al., 2025) further relaxed training restrictions, yielding more effective speculative decoding.

**Batching Method** A complementary line of work studies how batching can be combined with speculative decoding to better leverage GPUs. Existing methods mainly target the conventional chain-based paradigm, while tree-structured batching remains largely unexplored.

Su et al. (2023) first analyzed how batch size affects chain-style decoding, revealing trade-offs between improved parallelism and synchronization overhead. Building on this, Qian et al. (2024) proposed a strategy that parallelizes not only across batches but also along the draft-token axis, enabling finer GPU utilization and higher throughput. Most recently, Wu et al. (2025) introduced specialized techniques that further boost batched speculative decoding, demonstrating that careful batching design can accelerate inference at scale.

## 3 PRELIMINARY

In this section, we will briefly recap some of the needed knowledge and notions in LLM inference. Let $x_{1:t} = (x_1, x_2, \ldots, x_t)$ denote the language sequence. We will consider two autoregressive models as follows:

- **Target Model:** $P_T(x_{t+1} \mid x_{1:t})$, the high-quality, accurate model whose predictions we aim to approximate efficiently, which is usually a large model and has a bigger inference cost.
- **Draft Model:** $P_D(x_{t+1} \mid x_{1:t})$, a lightweight, fast model used to propose candidate tokens.

The objective is to sample from $P_T$ more efficiently using $P_D$ without compromising the quality of the output distribution.

### 3.1 SPECULATIVE DECODING

The motivation of speculative decoding (Leviathan et al., 2023; Chen et al., 2023) is that some tokens may be "easy" to predict and can use a smaller model to generate to make inference more efficient, and also the initial model is used to verify the correctness of the predictions.

Given context $x_{1:t}$, the draft model will first generate a sequence of $d$ tokens autoregressive: $\hat{x}_{t+1} \sim P_D(\cdot \mid x_{1:t})$, $\hat{x}_{t+2} \sim P_D(\cdot \mid x_{1:t}, \hat{x}_{t+1}) \cdots \hat{x}_{t+d} \sim P_D(\cdot \mid x_{1:t}, \hat{x}_{t+1:t+d-1})$. Let $\hat{x}_{t+1:t+d}$ denote the predicted draft sequence, the tokens are verified sequentially and once a token is accepted by the target model, we can drop the hat symbol.

Starting from $i = 1$. each token $\hat{x}_{t+i}$ is verified by the target model as follows:

- Calculate the draft probability: $q_i := P_D(\hat{x}_{t+i} \mid x_{1:t+i-1})$.
- Calculate the target probability: $p_i := P_T(\hat{x}_{t+i} \mid x_{1:t+i-1})$.

A uniform random number $u_i \sim \text{Uniform}(0, 1)$ is drawn. The token is accepted if: $u_i \leq \min\left(1, \frac{p_i}{q_i}\right)$. Otherwise, we reject the remaining tokens and fall back to sampling from a residual distribution: $x_{t+i} \sim \tilde{P}_T(\cdot \mid x_{1:t+i-1})$. where $\tilde{P}_T = \text{norm}(\max(0, p - q))$.

It can be shown that the above procedures can ensure the overall output sequence is sampled from the target model distribution $P_T$ (Leviathan et al., 2023).

### 3.2 EAGLE

The previously discussed speculative decoding method predicts the tokens in an autoregressive chain and verifies them sequentially. It has the disadvantage of once rejecting a token, all its subsequent tokens will also be discarded. EAGLE (Li et al., 2024b) improves speculative decoding by constructing a tree-structured draft and performing parallel verification, the tree structure makes the rejection process still retain some information by leaving the tokens in the rejected token's sibling subtree un-discarded.

Unlike EAGLE (Li et al., 2024b), which uses a predefined static tree, EAGLE-2 (Li et al., 2024c) and EAGLE-3 (Li et al., 2025) improve speculative decoding by dynamically constructing a tree-structured draft using the confidence score. The dynamic structure makes the inference much more data-dependent and performs much better. We will then briefly discuss some of the details.

### 3.2.1 TREE EXPANSION PHASE

To organize the token sequence into a tree structure, one may have the following two definitions:

- Each node $u$ corresponds to a token $x_u$ and its preceding context $c_u$.
- The (confidence) value of a node is:

$$v(u) = \prod_{w \in \text{path}(u)} P_D(x_w \mid c_w),$$

representing the confidence score by traveling along the draft path, and the root node will have a probability of 1.

Starting from the root (initial context) node, EAGLE-2 (and EAGLE-3) dynamically expands the draft tree layer by layer, and the tree will be of depth $H$:

1. At each level except the last layer, select top-$K$ nodes with the highest $v(u)$.
2. For each selected node $u$, generate $K$ child nodes by sampling from $P_D(\cdot \mid c_u)$.

### 3.2.2 TREE RERANKING PHASE

In the expansion stage, the goal is to further develop the draft tree by exploring deeper paths. However, because node values—interpreted as acceptance probabilities—lie between 0 and 1, they naturally diminish with depth. To address this, a reranking over all candidate tokens will be performed, and the top $m$ tokens with the highest associated values will be selected. An important constraint is that the value assigned to any node does not exceed that of its parent. Therefore, after reranking, it still comprises a valid subtree within the original draft structure.

After selection, the subtree will be linearized into a flat sequence to produce the input for the verification stage. To maintain compatibility with standard autoregressive decoding, the attention mask will also be changed. Consequently, the attention mask is modified such that each token attends only to its ancestors, preserving the hierarchical dependencies encoded in the tree.

## 4 METHOD

Though EAGLE-2 (EAGLE-3) has constructed a dynamic tree to increase the inference performance, its construction rule is mostly based on heuristics and does not consider the intricate interplay of the inference algorithm and GPU hardware, especially in the case of batched processing. When using batching techniques, merely increasing the tree depth and node numbers may not always result in better performance. This is because the GPU utilization has already increased by using batching, and naively adopting the speculative decoding methods may result in competition in the GPU resources and slow down the process.

Therefore, we should also consider the cost of inference during speculative decoding. Given a batch of $B$ samples, each with a context of length $c$, the inference time of inputting a length $n$ sequence will depends on $B, c, n$, which is denoted as $f(B, c, n)$. To save the time of inference, we can precompute the time and maintain a lookup table. To save the computation and storage, we only need to maintain the data of $f(B, c, n)$ for $c = kL$ ($k = 1, \cdots, M$) and $n = 1, \cdots, N$. And also the associated select operator $\text{select}(c) = (\max(\lfloor \frac{c}{L} \rfloor, M - 1) + 1)L$.

Then, for each needed size $B$, one can maintain the following two lookup tables:

$$S_T(B) = \{f_T(B, c, n)\} \quad \text{and} \quad S_D(B) = \{f_D(B, c, n)\},$$

where $f_T$ is for target model and $f_D$ for draft model. For a given context length $c$, $S_D(B)[\text{select}(c)]$ will return an array of size $N$.

Given a batch of $B$ samples, w.l.o.g. we can assume they have the same context length $n_0$ thanks to the padding technique and denote them as $x_{1:n_0}^j$ ($j = 1, \cdots, B$). As EAGLE has two stages, namely the expansion stage and reranking stage, when constructing the draft tree, we will tackle these two one by one.

## 4.1 DYNAMIC EXPANSION STAGE: BREADTH AND DEPTH PRUNING

The expansion stage of the draft tree construction involves two key dimensions: (1) the number of nodes per layer, and (2) the total number of layers in the tree. These two components are inherently coupled. An illustrative example can be found in Figure 2. We first focus on determining the number of nodes to retain in each layer, a process we refer to as *breadth pruning*.

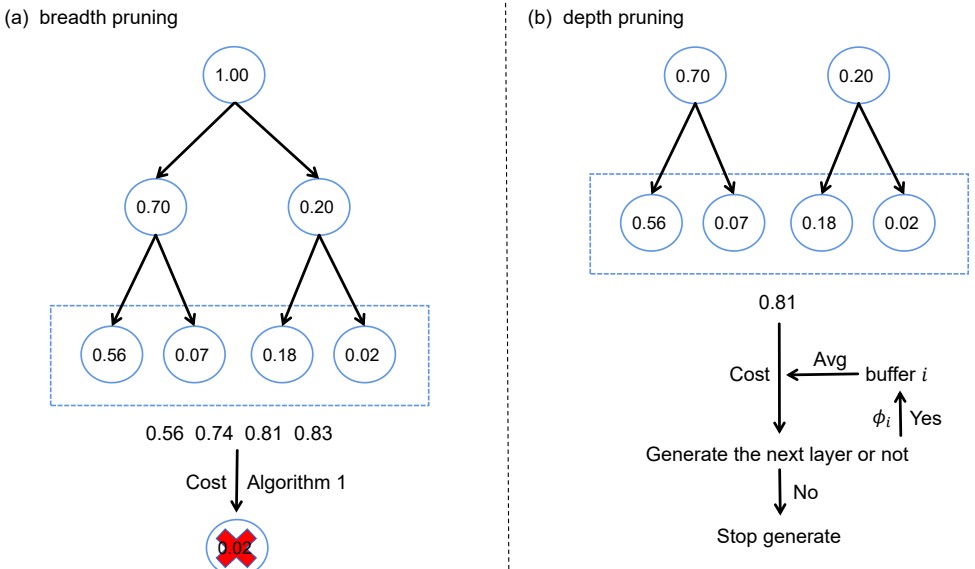

Figure 1: An illustrative example for the dynamic expansion stage, we use batch size as 1 for simplicity, general cases are tackled by averaging along batches. Each node will initially have 2 branches in the example.

The primary objective is to minimize the average inference latency per sequence. To this end, we aim to select draft tokens that are highly likely to be accepted by the target model. However, predicting an excessive number of tokens can increase overall latency, due to the additional computational cost. Thus, a tradeoff must be considered between the likelihood of token acceptance and the cost of incurring new predictions.

Empirically, the acceptance rate of a node $u$ is strongly correlated with its confidence score $v(u)$ (Du et al., 2024; Li et al., 2024c), which we use as a proxy for acceptance probability. Drawing inspiration from utility theory in economics, we frame node selection as a utility maximization problem.

Specifically, for the $i$-th layer, we denote the confidence scores of the $N_i$ candidate nodes (sorted in descending order) for each sample $j \in \{1, \ldots, B\}$ as $v_i^{(j)}(s)$, where $s \in \{1, \ldots, N_i\}$ and $v_i^{(j)}(1) \geq \cdots \geq v_i^{(j)}(N_i)$. The cumulative utility of selecting the top $k$ nodes is defined as:

$$u_k^{(i)} = \frac{1}{B} \sum_{j=1}^{B} \sum_{s=1}^{k} v_i^{(j)}(s). \tag{1}$$

Let $n_j$ denote the number of nodes retained in layer $j$, for $j = 1, \ldots, i-1$. The context length for layer $i$ is then $\sum_{j=1}^{i-1} n_j$. The normalized cost of selecting $k$ nodes at layer $i$, using the draft model relative to the target model cost, is computed as:

$$c_k^{(i)} = \frac{S_D(B)[\text{select}(\sum_{j=1}^{i-1} n_j)][k]}{S_T(B)[\text{select}(\sum_{j=1}^{i-1} n_j)][1]}. \tag{2}$$

In economic theory, utility functions are typically concave, exhibiting diminishing marginal utility. For a concave function $u(c)$ defined on $\mathbb{R}_+$, the marginal utility $\frac{u(c)-u(c_0)}{c-c_0}$ decreases as $c$ increases. Based on this principle, we introduce a threshold $C_1$ and retain nodes whose marginal utility exceeds

this threshold. The intial number of nodes to be chosen in each layer will be determined by the top-K probability in the previous layer, similar to EAGLE.

Due to the discrete nature of our setting, the utility function may not be strictly concave. A robust selection strategy is summarized in Algorithm 1, which takes as input the utility sequence $\{u_k^{(i)}\}$, the associated cost sequence $\{c_k^{(i)}\}$, and the threshold $C_1$, to determine the number of nodes $n_i$ to retain at layer $i$. Notably, the node selection mechanisms in EAGLE-2 and EAGLE-3 can be viewed as special cases of this generalized formulation.

**Theorem 4.1.** *EAGLE-2 and EAGLE-3's selection algorithm in $i$-th layer is a special case of the proposed selection Algorithm by setting $c_j = \lambda j + \delta$ and $C_1 = \frac{\sum_{j=1}^{B} v_i^{(j)}(K)}{B\lambda}$.*

---

**Algorithm 1** Select Maximum Valid Index

---

1: **Input:** Arrays $u[1\ldots n]$, $c[1\ldots n]$ strictly increasing; constant $C > 0$
2: Initialize $mark[1\ldots n] \leftarrow 1$
3: **for** $i = 1$ to $n$ **do**
4:     **for** $j = i + 1$ to $n$ **do**
5:         **if** $\dfrac{u[j] - u[i]}{c[j] - c[i]} < C$ **then**
6:             $mark[j] \leftarrow 0$
7:         **end if**
8:     **end for**
9: **end for**
10: **Output:** $\max\{j \mid mark[j] = 1\}$

---

Next, we consider *depth pruning*, which determines whether an additional layer $(i + 1)$ should be generated. This decision is based on the predictive relationship between successive layers. Let $\mathcal{A}_i$ be a buffer that tracks predictive quality for layer $i$. We define: $\alpha_i = \mathrm{Avg}(\mathcal{A}_i)$, where $\mathrm{Avg}$ denotes the average over the elements in $\mathcal{A}_i$. We proceed to generate layer $(i + 1)$ only if the following condition holds: $\alpha_i \cdot \frac{u_{n_i}^{(i)}}{c_{n_i}^{(i)}} \geq C_2$, where $C_2$ is a predefined threshold. Once this condition is satisfied and the number of nodes $n_{i+1}$ has been determined via breadth pruning, we compute the confidence gain ratio: $\phi_i = \frac{u_{n_{i+1}}^{(i+1)}}{u_{n_i}^{(i)}}$. We then update the buffer $\mathcal{A}_i$ using a first-in-first-out (FIFO) policy, maintaining up to $R$ recent values of $\phi_i$. Each buffer $\mathcal{A}_i$ is initialized with the value $\{1\}$ to ensure stability in early layers.

## 4.2 DYNAMIC RERANKING STAGE

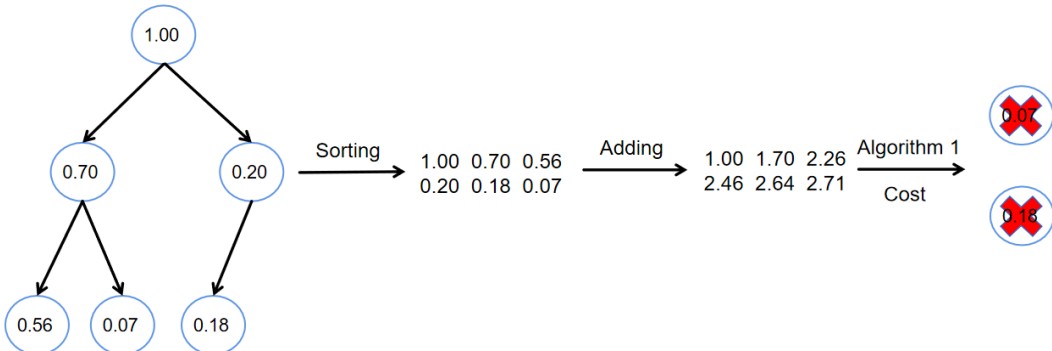

Figure 2: An illustrative example for the dynamic reranking stage.

After the dynamic expansion stage, a rooted draft tree is constructed, but with too many nodes that need to be further trimmed. We first consider collecting data samples and calculating each sample's

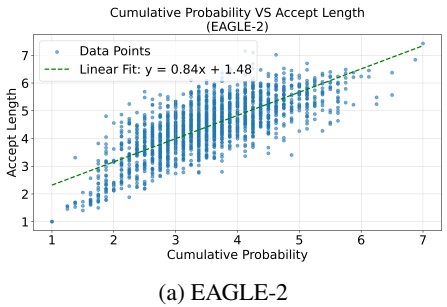

(a) EAGLE-2

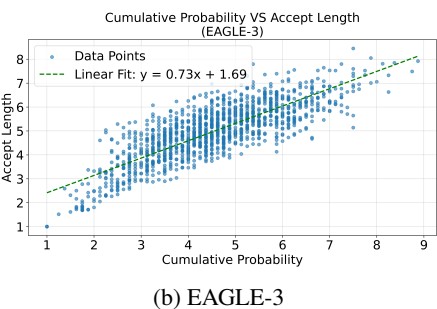

(b) EAGLE-3

Figure 3: The correlation of accept length and cumulative probability.

accept length and the cumulative probability score $v$ on the whole tree which is plotted in Figure 3. From the Figure, it is clear that the accept length and the cumulative probability shares a linear trend. Therefore, in order to maximize the accept length of each sample, one should make the cumulative probability as big as possible. Thus, choosing the nodes with top probability score is the right choice. Suppose after the dynamic expansion stage, the (batch averaged) score on the whole tree is sorted as $v(1) \geq \cdots \geq v(N)$ ($N$ is the minimum of $\sum_{i=0}^{H} n_i$ and a predefined hyperparameter $m$). By taking the inference cost into account, one can also use Algorithm 1 to determine the number of nodes to be verified by the target model by setting $u_k = \sum_{j=1}^{k} v(j)$ and $c_k = \frac{S_T(B)[\text{select}(n_0)][k]}{S_T(B)[\text{select}(n_0)][1]}$ with threshold constant $C_3$.

## 5 EXPERIMENTS

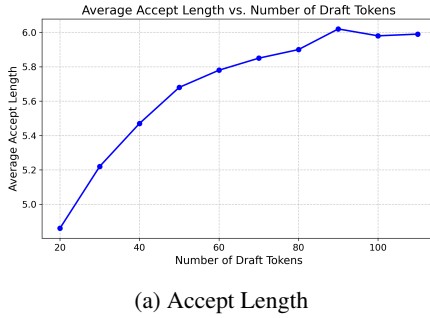

(a) Accept Length

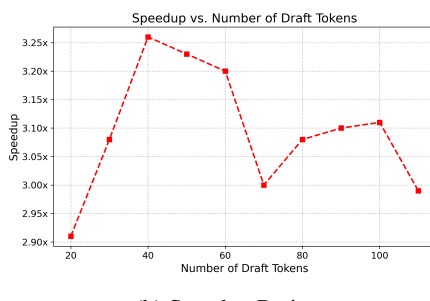

(b) Speedup Ratio

Figure 4: The behavior of accept length and speedup ratio when varying the number of tokens to be verified by the target model using EAGLE-3.

Following prior works, we perform experiments on a variety of models across diverse sizes, including Vicuna-13B-v1.3 (V 13B) and Vicuna-33B-v1.3(V 33B)(Chiang et al., 2023),Llama-3.1-8B-Instruct(L31 8B) and Llama-3.3-70B-Instruct(L33 70B) (Meta, 2024), DeepSeek-R1-Distill-Llama-8B(DSL 8B), Qwen2-7B-Instruct(Q2 7B). Following the standard benchmark in this area, we conducted extensive evaluations across six different text generation tasks to show the applicability of our method under diverse scenarios, includng multi-turn conversation, code generation, mathematical reasoning, instruction following, summarization, and question answering, we used the MT-bench (Zheng et al., 2023), HumanEval (Chen et al., 2021), GSM8K (Cobbe et al., 2021), Alpaca (Taori et al., 2023), CNN/Daily Mail (Nallapati et al., 2016), and Natural Questions (Kwiatkowski et al., 2019) datasets, respectively. In line with common practices in the community, we employed the same initial model weights for all tasks without any modifications. We will use the vanilla autoregressive decoding as the baseline for comparison, with a speedup ratio of 1.00x. Our method will be compared with the most state-of-the-art methods in speculative decoding and we will use their default hyperparameters, including standard speculative decoding (SpD) (Joao Gante, 2023; Leviathan et al., 2023; Chen et al., 2023), Medusa (Cai et al., 2024), PLD (Saxena, 2023), Lookahead (Fu et al., 2023), EAGLE (Li et al., 2024b), EAGLE-2 (Li et al., 2024c), and EAGLE-3 (Li et al., 2025). Under

non-greedy setting, methods like Medusa relax acceptance condition, so we will not compare with method like this. Given that speedup ratios are hardware-dependent, we ensured fairness by testing all methods on identical devices, which are the Nvidia A800 GPUs. All experiments run relatively fast, usually less than one hour, even for large datasets, as only inference is performed. The experiments on more GPU types can be found in Appendix.

As we mainly consider the *lossless* acceleration technique that neither fine-tunes the original LLM nor alters its acceptance conditions. As a result, we focus on evaluating its acceleration performance using the following metric. **Speedup Ratio:** The actual increase in speed compared to standard autoregressive decoding in a single run and verification round.

We do not adopt the metric Average Acceptance Length (The average number of tokens generated per drafting-verification cycle, indicating how many tokens are accepted from the draft.). This is because this metric may be somewhat misleading, particular in the larger batch case. In Figure 4, one can see that as the maximum number of verified token number $m$ is increasing, the accept length is increasing but sacrifices speedup when $m$ is relatively larger.

## 5.1 SINGLE SAMPLE CASE

Table 1: Comparison of Model Performance (Speedup Ratios) when batch size is 1.

| Model | Method | MT-bench | HumanEval | GSM8K | Alpaca | CNN/DM | Natural Ques. |
|---|---|---|---|---|---|---|---|
| | | | | Temperature=0 | | | |
| V 13B | SpD | 1.93x | 2.23x | 1.77x | 1.76x | 1.93x | 1.66x |
| | PLD | 1.58x | 1.85x | 1.68x | 1.16x | 2.42x | 1.14x |
| | Medusa | 2.07x | 2.50x | 2.23x | 2.08x | 1.71x | 1.81x |
| | Lookahead | 1.65x | 1.71x | 1.81x | 1.46x | 1.46x | 1.36x |
| | EAGLE | 2.61x | 3.58x | 3.08x | 2.93x | 2.80x | 3.02x |
| | EAGLE-2 | 3.02x | 4.06x | 3.35x | 3.25x | 3.40x | 3.13x |
| | EAGLE-3 | 3.70x | 4.73x | **4.00x** | **3.86x** | 3.68x | 3.31x |
| | CAST (Ours) | **3.98x** | **5.18x** | 3.98x | 3.80x | **3.76x** | **3.40x** |
| L33 70B | EAGLE-3 | 4.13x | 4.98x | 4.63x | 4.66x | 3.50x | 3.61x |
| | CAST (Ours) | **4.23x** | **5.23x** | **4.65x** | **4.83x** | **3.56x** | **3.67x** |
| L31 8B | EAGLE-3 | 3.60x | 4.27x | 3.82x | 4.00x | 3.22x | 3.06x |
| | CAST (Ours) | **3.77x** | **4.51x** | **3.95x** | **3.98x** | **3.32x** | **3.22x** |
| DSL 8B | EAGLE-3 | 3.47x | 3.78x | 3.68x | 3.20x | 2.90x | 2.95x |
| | CAST (Ours) | **3.63x** | **3.85x** | **3.98x** | **3.37x** | **3.02x** | **3.20x** |
| | | | | Temperature=1 | | | |
| V 13B | SpD | 1.62x | 1.72x | 1.46x | 1.52x | 1.66x | 1.43x |
| | EAGLE | 2.42x | 2.75x | 2.37x | 2.43x | 2.34x | 2.04x |
| | EAGLE-2 | 2.80x | 3.22x | 2.79x | 2.71x | 2.65x | 2.27x |
| | EAGLE-3 | 3.28x | 3.94x | 3.39x | 3.25x | 3.23x | 2.74x |
| | CAST (Ours) | **3.51x** | **4.30x** | **3.76x** | **3.38x** | **3.32x** | **2.95x** |
| L33 70B | EAGLE-3 | 3.96x | 4.73x | 4.37x | 4.39x | 3.42x | 3.50x |
| | CAST (Ours) | **4.19x** | **4.93x** | **4.51x** | **4.66x** | **3.50x** | **3.50x** |
| L31 8B | EAGLE-3 | 2.77x | 3.58x | 3.05x | 3.26x | 2.57x | 2.32x |
| | CAST (Ours) | **3.06x** | **3.91x** | **3.36x** | **3.41x** | **2.89x** | **2.53x** |
| DSL 8B | EAGLE-3 | 2.58x | 3.15x | 2.76x | 2.42x | 2.21x | 2.37x |
| | CAST (Ours) | **2.82x** | **3.43x** | **2.99x** | **2.65x** | **2.48x** | **2.66x** |

We begin our analysis by examining the usual setting in the literatures, namely when the batch size is 1. We term the proposed method as Cost-Aware Speculative Tree (CAST). To ensure a fair and rigorous comparison with existing methods, we adopt the same target model configuration used in comparision with the respective SOTA EAGLE family models. This alignment in experimental setup allows us to attribute any observed performance differences solely to the algorithmic innovations of CAST, rather than to variations in model size, training regime, or evaluation protocol.

The quantitative results are summarized in Table 1, which reports the speedup ratios achieved by CAST relative to prior baselines. As the table indicates, CAST usually yields higher speedup ratios across multiple evaluation tasks, underscoring its ability to more effectively utilize computational resources. This trend becomes increasingly evident as the size of the target model grows, suggesting

that our method scales particularly well in large-model scenarios where efficiency considerations are most critical. The advantage of CAST is especially striking on the HumanEval benchmark, where a speedup of 5.23 is achieved. These results collectively highlight the potential of our method as a practical solution for accelerating speculative decoding pipelines, particularly in demanding real-world settings where inference latency and throughput remain key bottlenecks.

## 5.2 BATCHING CASE

Table 2: Comparison of different methods across models and benchmarks when batch size is 8. All values are speedup ratios.

| Model | Method | MT-Bench | HumanEval | GSM8K | Alpaca | CNN/DM | Natural Ques. |
|---|---|---|---|---|---|---|---|
| | | | | Temperature=0 | | | |
| Q2 7B | EAGLE | 1.18x | 1.62x | 1.76x | 1.80x | 0.84x | 1.44x |
| | EAGLE-2 | 1.25x | 1.49x | 1.40x | 1.48x | 1.11x | 1.10x |
| | CAST (Ours) | **1.86x** | **2.16x** | **2.19x** | **2.06x** | **1.70x** | **1.72x** |
| L31 8B | EAGLE | 1.80x | 2.14x | 2.10x | 2.09x | 1.38x | 1.76x |
| | EAGLE-2 | 1.39x | 1.60x | 1.59x | 1.63x | 1.03x | 1.32x |
| | EAGLE-3 | 1.72x | 1.97x | 1.92x | 2.16x | 1.34x | 1.72x |
| | CAST (Ours) | **2.16x** | **2.62x** | **2.41x** | **2.62x** | **1.76x** | **2.11x** |
| V 13B | EAGLE | 1.63x | 1.91x | 1.79x | 1.72x | 1.37x | 1.51x |
| | EAGLE-2 | 1.25x | 1.42x | 1.30x | 1.28x | 1.02x | 1.03x |
| | EAGLE-3 | 1.59x | 1.91x | 1.67x | 1.80x | 1.37x | 1.39x |
| | CAST (Ours) | **2.48x** | **3.12x** | **2.61x** | **2.76x** | **1.97x** | **2.27x** |
| V 33B | EAGLE | 1.78x | 2.09x | 1.96x | 1.75x | 1.44x | 1.47x |
| | EAGLE-2 | 1.27x | 1.50x | 1.37x | 1.26x | 1.05x | 1.01x |
| | CAST (Ours) | **2.12x** | **2.48x** | **2.21x** | **2.09x** | **1.79x** | **1.84x** |
| | | | | Temperature=1 | | | |
| Q2 7B | EAGLE | 0.80x | 1.06x | 1.21x | 1.15x | 0.62x | 1.00x |
| | EAGLE-2 | 0.93x | 1.27x | 1.30x | 1.16x | 0.83x | 0.92x |
| | CAST (Ours) | **1.50x** | **1.96x** | **1.94x** | **1.82x** | **1.40x** | **1.57x** |
| L31 8B | EAGLE | 1.24x | 1.53x | 1.47x | 1.57x | 1.06x | 1.23x |
| | EAGLE-2 | 1.07x | 1.48x | 1.39x | 1.47x | 0.93x | 1.07x |
| | EAGLE-3 | 1.25x | 1.70x | 1.67x | 1.90x | 1.13x | 1.32x |
| | CAST (Ours) | **1.73x** | **2.37x** | **2.26x** | **2.46x** | **1.69x** | **1.76x** |
| V 13B | EAGLE | 1.25x | 1.39x | 1.39x | 1.34x | 1.10x | 1.11x |
| | EAGLE-2 | 1.14x | 1.22x | 1.22x | 1.11x | 0.94x | 0.95x |
| | EAGLE-3 | 1.28x | 1.56x | 1.45x | 1.34x | 1.18x | 1.28x |
| | CAST (Ours) | **2.08x** | **2.51x** | **2.22x** | **2.11x** | **1.77x** | **2.16x** |
| V 33B | EAGLE | 1.48x | 1.66x | 1.64x | 1.49x | 1.20x | 1.26x |
| | EAGLE-2 | 1.18x | 1.37x | 1.34x | 1.14x | 1.01x | 0.97x |
| | CAST (Ours) | **1.97x** | **2.16x** | **2.11x** | **1.95x** | **1.68x** | **1.79x** |

When moving beyond the single-sample setting to scenarios where multiple samples are processed simultaneously, batching becomes a crucial factor in evaluating the practicality of speculative decoding methods. In this regime, our study primarily focuses on comparisons with SOTA tree-based speculative decoding approaches, which represent the most competitive baselines in this line of research. Table 2 provides a comprehensive evaluation of CAST against these baselines under the batching setting where the batch size is fixed at 8. The evaluation spans a diverse collection of LLMs, benchmark tasks, and decoding temperatures, ensuring that the reported results reflect a broad and robust performance profile rather than being limited to a narrow set of conditions.

The empirical results reveal a clear and consistent advantage for CAST across the tested scenarios. Specifically, CAST achieves speedups of up to 3.12x in challenging tasks such as V13B-HumanEval at temperature 0, and up to 2.51x in V13B-MT-Bench at temperature 1. The results show the potential of our method under the batching cases. On average, CAST achieves relative improvements in the range of 5% to 20%, reflecting tangible efficiency gains without compromising correctness.

## 6 CONCLUSION

In this work, we present a cost-aware dynamic tree-based speculative decoding method that adapts to system-level factors such as device type and batch size. By modeling the trade-off between accept length and inference speed, our method CAST dynamically adjusts the draft tree structure for more efficient decoding. Extensive experiments across diverse tasks and models demonstrate that our approach generally outperforms prior methods, achieving up to $5.2$ speedup and $5 - 20\%$ efficiency gains over the best baselines.

## ACKNOWLEDGMENTS

This work was supported by the Science and Technology Major Project of Yunnan Province (202302AF080006), the National Key R&D Program of China (2022YFB2703500).

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

# Appendix

## A    ETHICS STATEMENT

This work adheres to the ICLR Code of Ethics. In this study, no human subjects or animal experimentation was involved. All datasets used, including MT-bench, HumanEval, GSM8K, Alpaca, CNN/Daily Mail, and Natural Questions, were sourced in compliance with relevant usage guidelines, ensuring no violation of privacy. We have taken care to avoid any biases or discriminatory outcomes in our research process. No personally identifiable information was used, and no experiments were conducted that could raise privacy or security concerns. We are committed to maintaining transparency and integrity throughout the research process.

## B    REPRODUCIBILITY STATEMENT

We have made every effort to ensure that the results presented in this paper are reproducible. The experimental setup, for example the model configurations, and hardware details, is described in detail in the paper. We have also provided a full description of the algorithm details, to assist others in reproducing our experiments.

Additionally, public datasets used in the paper, such as MT-bench, HumanEval, GSM8K, Alpaca, CNN/Daily Mail, and Natural Questions, are publicly available, ensuring consistent and reproducible evaluation results.

We believe these measures will enable other researchers to reproduce our work and further advance the field.

## C    LLM USAGE

Large Language Models (LLMs) were used to aid in the polishing of the manuscript. Specifically, we used an LLM to assist in refining the language, improving readability, and ensuring clarity in various sections of the paper. The model helped with tasks such as sentence rephrasing, grammar checking, and enhancing the overall flow of the text.

It is important to note that the LLM was not involved in the ideation, research methodology, or experimental design. All research concepts, ideas, and analyses were developed and conducted by the authors. The contributions of the LLM were solely focused on improving the linguistic quality of the paper, with no involvement in the scientific content or data analysis.

The authors take full responsibility for the content of the manuscript, including any text generated or polished by the LLM. We have ensured that the LLM-generated text adheres to ethical guidelines and does not contribute to plagiarism or scientific misconduct.

## D    PROOF

**Theorem D.1.** *EAGLE-2 and EAGLE-3's selection algorithm in $i$-th layer is a special case of the proposed selection Algorithm by setting $c_j = \lambda j + \delta$ and $C_1 = \frac{\sum_{j=1}^{B} v_i^{(j)}(K)}{B\lambda}$.*

*Proof.* Note $v_i^{(j)}(s)$ is decreasing about $s$ and $u$ is constructed by prefix sum. Then we know $\max_{j>k} \frac{u[j]-u[k]}{c[j]-c[k]} = \max_{j>k} \frac{1}{\lambda} \frac{u[j]-u[k]}{j-k} = \frac{u[k+1]-u[k]}{\lambda} = \frac{\sum_{j=1}^{B} v_i^{(j)}(k+1)}{B\lambda}$. By also noticing that the mean of a sequence is larger than its minimum, the maximum non-zero index will be $K$. $\qquad\square$

## E    MORE IMPLEMENTATION DETAILS

In this section, we will present more details of our implementation. And our method may have the potential limitation of pecomputing the inference cost.

In experiments conducted with a batch size of 1:

- The Llama-3.3-70B-Instruct and Vicuna-13B-v1.3 models utilized a threshold of 4.
- The Llama-3.1-8B-Instruct and DeepSeek-R1-Distill-Llama-8B models utilized a threshold of 3.
- The Llama-3.3-70B-Instruct model was run in a dual-card environment (2x A800 GPUs), while the other three models were run in a single-card environment (1x A800 GPU).
- For our improved algorithm, all models used the following parameters: depth=13, total_token=72, and top_k=12.
- EAGLE-3 employed its default parameters, namely depth=7 and top_k=10, with `total_token` configured according to the specific model (refer to the appendix of the EAGLE-3 paper for details).

In experiments conducted with a batch size of 8:

- We utilized a single-card A800 GPU environment.
- For our method, we uniformly applied a threshold of 2.5, a depth of 9, a top_k of 12, and a `total_token` count of 72.
- For comparison, EAGLE, EAGLE-2, and EAGLE-3 were configured with their respective default parameters.

For the ablation studies:

- Due to the involvement of large batch sizes, all experiments were conducted in a dual-card environment (2x A800 GPUs).
- It is important to note that speedup ratios measured in single-card versus dual-card environments can exhibit a little difference.
- For more comprehensive hyperparameter settings, including specific values for each parameter and detailed reproduction methodologies, please consult our supplementary materials.

### E.1 THE EFFECT OF BATCH SIZE

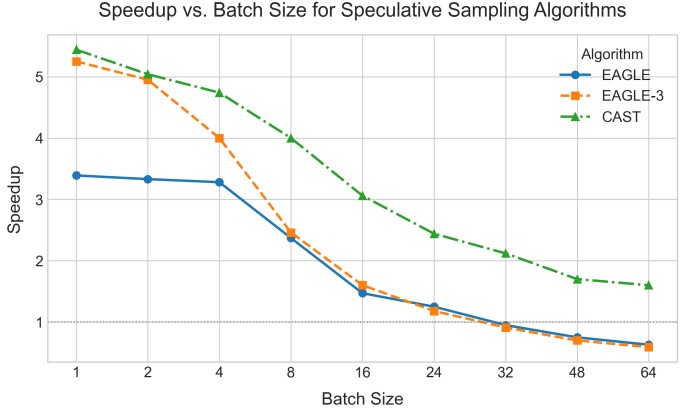

Figure 5: The speedup under different batch sizes on HumanEval.

Figure 5 presents a comparative analysis of speedup achieved by three speculative decoding algorithms under various batch sizes—EAGLE, EAGLE-3, and CAST—as a function of batch size. We observe that CAST consistently yields the highest speedup across all batch sizes, demonstrating strong scalability and robustness to increasing batch sizes. It achieves a peak speedup exceeding 5x at batch size 1 and maintains over 2x speedup even at batch size 32. EAGLE-3 shows moderate performance, outperforming EAGLE at smaller batch sizes but converging toward similar performance as batch size increases. EAGLE, while providing stable gains at small batch sizes (e.g., 3.4x speedup at batch

size 1), suffers from a rapid drop in efficiency as batch size grows, eventually offering marginal speedup (close to 1x) beyond batch size 32.

This trend illustrates a key limitation of baseline speculative decoding under large-batch settings and highlights the effectiveness of CAST in mitigating this degradation. The improved performance of CAST is attributed to its enhanced speculative mechanism, which more accurately predicts and validates multiple tokens in parallel, thus reducing the need for fallback to the base model.

### E.2 THE EFFECT OF EACH COMPONENT OF CAST

Table 3 presents the results of an ablation study on CAST, our enhanced speculative decoding algorithm, which extends EAGLE-3 by progressively integrating three key optimization techniques: Dynamic Reranking (DR), Depth Pruning (DP), and Breadth Pruning (BP). The baseline EAGLE-3 demonstrates strong initial performance but degrades significantly as batch size increases, falling to 1.35x at batch size 16. Adding DR alone yields slight gains at larger batch sizes (e.g., 2.17x at batch size 16), while incorporating DP further improves performance consistently across batch sizes. The combination of DR + DP + BP (i.e., the full CAST system) achieves the best overall speedups, culminating in a 4.14x speedup at batch size 1 and maintaining a robust 2.35x speedup at batch size 16. Notably, each additional component contributes marginal gains, confirming the cumulative effectiveness of the enhancements.

Table 3: Ablation study of CAST components.

| Batch size | 1 | 2 | 4 | 8 | 16 |
|---|---|---|---|---|---|
| EAGLE3 | 3.99x | 3.79x | 2.98x | 1.91x | 1.35x |
| EAGLE3+DR | 3.99x | 3.74x | 3.44x | 2.77x | 2.17x |
| EAGLE3+DR+DP | 4.08x | 3.82x | 3.44x | 2.84x | 2.27x |
| EAGLE3+DR+BP | 4.06x | 3.80x | 3.42x | 2.79x | 2.26x |
| CAST | 4.14x | 3.87x | 3.48x | 2.91x | 2.35x |

## F MORE RESULTS ON DIFFERENT GPUS

We present more experimental results on H20 and 4090 to show the flexibility of our methods on different GPU devices.

Table 4: Performance comparison of EAGLE-3 and CAST (Ours) on a single H20 GPU with Batch Size 1. Values represent speedup factors.

| Model | Method | MT-Bench | HumanEval | GSM8K | Alpaca | CNN/DM | Natural Ques. |
|---|---|---|---|---|---|---|---|
| | | | | Temperature=0 | | | |
| L33 70B | EAGLE-3 | 5.28x | 6.46x | 5.86x | 5.80x | 4.28x | 4.59x |
| | CAST (Ours) | **5.40x** | **6.66x** | **5.86x** | **5.95x** | **4.35x** | **4.71x** |
| V 13B | EAGLE-3 | 3.13x | 3.76x | 3.14x | 3.22x | 2.85x | 2.65x |
| | CAST (Ours) | **3.30x** | **4.27x** | **3.26x** | **3.25x** | **2.95x** | **2.73x** |
| L31 8B | EAGLE-3 | 3.57x | 3.94x | 3.67x | 3.84x | 3.04x | 2.99x |
| | CAST (Ours) | **3.64x** | **4.24x** | **3.68x** | **3.90x** | **3.19x** | **3.16x** |
| DSL 8B | EAGLE-3 | 3.34x | 3.95x | 3.83x | 3.11x | 2.82x | 2.99x |
| | CAST (Ours) | **3.50x** | **3.95x** | **4.13x** | **3.27x** | **2.96x** | **3.15x** |
| | | | | Temperature=1 | | | |
| L33 70B | EAGLE-3 | 5.09x | 6.09x | 5.56x | 5.52x | 4.17x | 4.50x |
| | CAST (Ours) | **5.21x** | **6.36x** | **5.68x** | **5.78x** | **4.26x** | **4.61x** |
| V 13B | EAGLE-3 | 2.54x | 3.25x | 2.62x | 2.65x | 2.53x | 2.33x |
| | CAST (Ours) | **2.92x** | **3.57x** | **2.89x** | **2.89x** | **2.62x** | **2.52x** |
| L31 8B | EAGLE-3 | 2.71x | 3.68x | 3.19x | **3.18x** | 2.51x | 2.28x |
| | CAST (Ours) | **2.87x** | **3.73x** | **3.22x** | 3.16x | **2.70x** | **2.45x** |
| DSL 8B | EAGLE-3 | 2.65x | **3.25x** | 2.83x | 2.46x | 2.15x | 2.31x |
| | CAST (Ours) | **2.78x** | 3.18x | **3.16x** | **2.67x** | **2.35x** | **2.59x** |

Table 5: Performance comparison of EAGLE-3 and CAST (Ours) on two RTX 4090 GPUs with Batch Size 1. Values represent speedup factors.

| Model | Method | MT-Bench | HumanEval | GSM8K | Alpaca | CNN/DM | Natural Ques. |
|---|---|---|---|---|---|---|---|
| | | | | Temperature=0 | | | |
| V 13B | EAGLE-3 | 4.28x | 5.02x | 4.17x | 4.06x | 3.90x | 3.35x |
| | CAST (Ours) | **4.54x** | **5.56x** | **4.38x** | **4.26x** | **4.07x** | **3.43x** |
| L31 8B | EAGLE-3 | 3.83x | 4.34x | 3.98x | 4.12x | 3.34x | 3.20x |
| | CAST (Ours) | **3.97x** | **4.61x** | **4.08x** | **4.29x** | **3.40x** | **3.32x** |
| DSL 8B | EAGLE-3 | 3.68x | 4.11x | 4.07x | 3.31x | 3.04x | 3.10x |
| | CAST (Ours) | **3.75x** | **4.15x** | **4.13x** | **3.43x** | **3.19x** | **3.22x** |
| | | | | Temperature=1 | | | |
| V 13B | EAGLE-3 | 3.48x | 3.89x | 3.47x | 3.44x | 3.51x | 3.12x |
| | CAST (Ours) | **3.78x** | **4.49x** | **3.90x** | **3.68x** | **3.60x** | **3.30x** |
| L31 8B | EAGLE-3 | 2.81x | 3.82x | 3.32x | 3.37x | 2.63x | 2.41x |
| | CAST (Ours) | **3.14x** | **3.99x** | **3.55x** | **3.55x** | **2.92x** | **2.61x** |
| DSL 8B | EAGLE-3 | 2.66x | 3.38x | 3.08x | 2.50x | 2.37x | 2.46x |
| | CAST (Ours) | **2.85x** | **3.52x** | **3.16x** | **2.72x** | **2.57x** | **2.69x** |

