# OpenReview forum: "Inference-Cost-Aware Dynamic Tree Construction for Efficient Inference in Large Language Models"
_ICLR.cc/2026/Conference — ICLR 2026 Poster_

### Official Review · Reviewer_Aptz · 2025-10-30

**Soundness:** 3
**Presentation:** 2
**Contribution:** 3
**Rating:** 4
**Confidence:** 2

**Summary:**

This paper addresses the latency challenge of autoregressive large language model inference by improving speculative decoding efficiency. Prior dynamic draft-tree methods such as EAGLE-2 and EAGLE-3 adapt token proposal structures, but they overlook system-level factors, including GPU characteristics and batch size. The proposed method models inference cost and dynamically adjusts tree depth, branching width, and token verification count to balance acceptance rates against computation overhead. Experiments demonstrate consistent gains and improvement over previous state-of-the-art methods.

**Strengths:**

- Adeaute related background along the timeline and solid motivation analysis.

- The paper explains its concepts clearly, and the writing is smooth and well-organised.

- The paper presents clear experimental settings and includes a comprehensive set of comparison models.

**Weaknesses:**

- For the conclution "merely increasing the tree depth and node numbers may not always result in better performance", is there some quantitative analysis / results to support?
- The results are mainly presented in table form. Is there a visualization of the performance trends? The paper also mentions a trade-off — are there additional experiments that explore multi-factor trade-offs in more depth? Just like Figure 5 in Appedndix E.1. Maybe more info in the Appendix could be included in the main body and cut off 1 table.
- 3.2.2 typo "valuesinterpreted"
- The core contribution of this paper lies in incorporating cost into the model’s decision process. However, the work does not provide concrete examples or detailed formulation for this component. Although the paper mentions that GPU hardware characteristics and batch size should be considered, it is necessary to further clarify the modeling procedure—for instance, which GPU parameters are included and how they quantitatively influence the cost function. This aspect is central to the method and requires empirical evidence to validate its effectiveness. Since the high-level idea is relatively straightforward, the design and justification of the cost function should serve as the main technical contribution.

**Questions:**

- General questions are given in the weakness part.

- Is there a theoretical upper bound or lower bound under specific settings? Furthermore, do the experimental results align with the theoretical analysis?

- Can you provide an example of how to represent hardware, such as parameters like memory capacity and FLOPS? Is there a unified quantitative representation for hardware characteristics?

---

> ### Author Response · Authors · 2025-11-19
>
> >Q1: For the conclusion "merely increasing the tree depth and node numbers may not always result in better performance," is there some quantitative analysis / results to support?
>
> A1: Yes, we have performed some quantitative results in Figure 4. As shown in Figure 4, the measured speedup exhibits a clear trend: performance initially improves as tree depth and node count increase, but eventually declines after exceeding a certain threshold. This empirical pattern directly supports our conclusion that deeper and larger trees do not always yield better results.
>
> >Q2: The results are mainly presented in table form. Is there a visualization of the performance trends? The paper also mentions a trade-off — are there additional experiments that explore multi-factor trade-offs in more depth? Just like Figure 5 in Appendix E.1. Maybe more info in the Appendix could be included in the main body and cut off 1 table.
>
> A2: Thank you for the helpful suggestion. In the final version of the paper, we plan to include additional performance-trend visualizations similar to Figure 5, and we will consider moving Figure 5 from the appendix into the main text since the camera-ready version allows for an extended page limit.
>
> Regarding multi-factor trade-offs, they are actually implicitly captured within our algorithm. Factors such as batch size, GPU type, multi-GPU usage will influence the cost distribution (i.e., the per-forward inference time under different values of batch size (B), context length (c), and verification count (n)). Since this cost distribution can be obtained through a one-time measurement procedure, our existing experiments already reflect the essential characteristics of the algorithm. Some results can be found in Tables 4 and 5 in the Appendix.
>
>
> >Q3: 3.2.2 typo "valuesinterpreted"
>
> A3: Thank you for pointing this out. We have corrected the typo and revised the paper accordingly.

---

> ### Author Response · Authors · 2025-11-19
>
> >Q4: The core contribution of this paper lies in incorporating cost into the model’s decision process. However, the work does not provide concrete examples or detailed formulation for this component. Although the paper mentions that GPU hardware characteristics and batch size should be considered, it is necessary to further clarify the modeling procedure—for instance, which GPU parameters are included and how they quantitatively influence the cost function. This aspect is central to the method and requires empirical evidence to validate its effectiveness. Since the high-level idea is relatively straightforward, the design and justification of the cost function should serve as the main technical contribution.
>
> A4: Due to space limitations, we were unable to fully elaborate on this part in the main text. We provide some additional clarification here, and we agree with your suggestion — we plan to include the detailed measurement methodology and related innovations in the final version of the paper, as it will have more page limit.
>
> In a fixed inference environment (e.g., a specific model such as Llama and a specific inference engine such as SGLang), our algorithm assumes that the per-forward inference time depends on three factors: context length (c), batch size (B), and verification count (n). Directly measuring all possible combinations would be expensive, so we introduce several optimizations.
>
> First, we discretize the context length (c) into intervals via a function (select(c)), with an interval size (L). Empirically, adjacent context lengths produce almost identical inference times. Thus, choosing (L=64) or (L=128) dramatically reduces measurement overhead. For example, with maximum context length (c=2048) and (L=64), only 32 interval measurements are required instead of 2048. In practice (SGLang + Llama3-8B), the intra-interval variation remains within ~1% for (L=64), and still sufficiently small even for (L=128).
>
> Second, we decompose inference time into two independent components: context-dependent operations (e.g., attention) and context-independent operations (e.g., FFN, quantization overhead, communication in distributed inference). We measure these two parts separately and sum them. This dramatically reduces measurement time and makes the overhead comparable across different model sizes.
>
> On SGLang using mainstream GPUs (RTX3090/4090, A100, H100), the total measurement time is on the order of tens of minutes. For example, on an H100 with Llama3-8B, the full measurement takes roughly 20–30 minutes. This is acceptable for production systems, which is the primary scenario our algorithm targets.
>
> >Q5: Is there a theoretical upper bound or lower bound under specific settings? Furthermore, do the experimental results align with the theoretical analysis?
>
> A5: Establishing strict theoretical upper or lower bounds in this setting is highly challenging. The performance of tree-based speculative decoding is fundamentally tied to the predictive quality of the draft model. Weak and strong draft models lead to drastically different acceptance rates and speedup behaviors, and—based on our understanding—the field currently lacks a theoretical framework capable of characterizing this variability. This reflects a broader limitation in deep learning, where many phenomena remain primarily empirical rather than theoretically grounded.
>
> For this reason, deriving formal bounds that remain valid across different draft–target model pairs is not feasible at present. This limitation applies not only to our method but also to prior tree-based speculative decoding approaches.
>
>
> >Q6: Can you provide an example of how to represent hardware, such as parameters like memory capacity and FLOPS? Is there a unified quantitative representation for hardware characteristics?
>
> A6: In our algorithm, explicitly quantifying hardware parameters is unnecessary. What matters is simply that different hardware and inference settings lead to different cost distributions, and our one-time cost measurement procedure already captures these differences. Thus, we do not need to know which specific hardware parameters caused a particular performance difference.
>
> If one insists on a unified quantitative representation of hardware, the most effective approach is precisely the empirical cost distribution: the per-forward inference time under different values of (B), (c), and (n). For example, in the same inference framework (e.g., SGLang, batch size (B = 8), context length (c = 1024), verification count (n = 32)), hardware A may yield 30 ms per forward step while hardware B yields 25 ms. This empirical distribution fully reflects the relevant hardware characteristics in a manner that is both practical and directly useful for our algorithm.

---

### Official Review · Reviewer_9P9F · 2025-10-31

**Soundness:** 2
**Presentation:** 2
**Contribution:** 2
**Rating:** 4
**Confidence:** 3

**Summary:**

This paper addresses the inference latency of llms by improving speculative decoding. The authors argue that existing dynamic tree-structured methods, such as EAGLE-2 and EAGLE-3, are suboptimal because they ignore system-level inference costs, particularly the impact of GPU hardware and batch size. This paper introduces Cost-Aware Speculative Tree, a new dynamic tree approach that explicitly models this trade-off. CAST uses pre-computed cost look-up tables to guide the dynamic construction and reranking of the draft token tree, pruning branches where the computational cost outweighs the expected utility. Experiments across six models and six tasks show that CAST achieves state-of-the-art speedups, outperforming prior methods by 5-20% and autoregressive decoding by up to 5.2x .

**Strengths:**

- It correctly identifies that SOTA dynamic tree methods ignore critical system costs like batch size and GPU type, which can negate speedups . The proposed cost-utility model, which uses precomputed lookup tables to guide tree construction, is an good solution.
- A key contribution is the generalization of prior SOTA (EAGLE-2/3), demonstrating they are special cases of this new framework (Theorem 4.1) . The empirical results are strong, showing consistent 5-20% gains over EAGLE-3 and demonstrating superior scalability as batch size increases—a vital metric for production systems.

**Weaknesses:**

- One of the weakness is the reliance on a new set of hyperparameters, specifically the cost thresholds $C_1$, $C_2$, and $C_3$ and the buffer size $R$, whose selection and sensitivity are not discussed or ablated.
- The method's practicality hinges on pre-computing cost-lookup tables $S_T(B)$ and $S_D(B)$. While practical, the paper does not sufficiently analyze the cost and complexity of this profiling step, which must be run for different hardware and batching configurations. It is unclear how the $select(c)$ approximation for context length impacts the cost model's accuracy.

**Questions:**

- How are the crucial thresholds $C_1, C_2, C_3$ and the FIFO buffer size $R$ determined? Please provide a sensitivity analysis for these new hyperparameters.
- What is the practical overhead (e.g., in hours) of generating the $S_T(B)$ and $S_D(B)$ lookup tables for a new model and hardware configuration? How sensitive is performance to the accuracy of this precomputed cost model?
- Algorithm 1 is used for both breadth pruning (Sec 4.1) and reranking (Sec 4.2), but with different cost functions ($c_k^{(i)}$ vs. $c_k$). Can you elaborate on the rationale for using the normalized draft cost for pruning but the normalized target cost for reranking?

---

> ### Author Response · Authors · 2025-11-19
>
> >Q1: One of the weakness is the reliance on a new set of hyperparameters, specifically the cost thresholds $C_{1} ,C_{2},C_{3}$ and the buffer size $R$, whose selection and sensitivity are not discussed or ablated.
>
> A1: In tree-based speculative sampling, a draft model iteratively generates a set of nodes, which are then pruned to form a draft tree, and finally validated in parallel by the target model. The core of our algorithm is the efficient pruning of this dynamic draft tree based on inference cost.
>
> 1.  Cost Thresholds ($C_{1}, C_{2}, C_{3}$): These thresholds are not set randomly but are determined based on the algorithm's principle of pruning nodes with poor cost-effectiveness (low speedup). They need to be representative of typical acceleration ratios. $C_{3}$ (Parallel Validation Pruning): This threshold is set to the average speedup observed over the entire inference process. This average can be reliably determined after running the system a few times (it typically ranges from $1.5$ to $3.5$; $2.5$ can be used as a safe initial estimate). $C_{3}$ dictates how many nodes are ultimately submitted for parallel validation by the target model. $C_{2}$ (Depth Pruning): $C_{2}$ is the speedup threshold for determining whether to perform another iteration of the draft model (controlling tree depth). It is set relative to $C_{3}$: ${C_{2} = 1.5 \times C_{3}}$. $C_{1}$ (Breadth Pruning): $C_{1}$ is the speedup threshold for determining the number of nodes to keep in a single draft model iteration (controlling tree width). It is set as: ${C_{1} = 2.0 \times C_{3}}$.
>
> **Sensitivity:** We found that the overall speedup remains quite **stable** as long as these acceleration ratio thresholds are within a reasonable range and do not deviate significantly from the average speedup. The approach is robust to minor variations in these settings. For example, for the Vicuna 13B on HumanEval, we will have the following table. From the table, we can see that it is not very sensitive to the choice of hyperparameters.
>
> | $C_1$   | Speedup |
> |-----|-------------|
> | 5   |        5.12x     |
> | 5.5 |       5.1x      |
> | 6   |     5.18x        |
> | 6.5 |       5.15x      |
>
>
>
> 2.  The determination of the buffer size $R$ is straightforward. $R$ is the size of the historical buffer used to calculate the average score loss ratio between adjacent layers during the depth pruning process (using $C_{2}$). This score loss ratio varies with the batch size. Smaller batches often keep most scores (loss ratio near $0.9$) due to wider trees, while larger batches prune more aggressively to save resources (loss ratio around $0.6$). We use the average score loss ratio from the most recent $R$ historical results. In practice, we typically use ${R=32}$.

---

> ### Author Response · Authors · 2025-11-19
>
> >Q2: The method's practicality hinges on pre-computing cost-lookup tables $S_{T} (B)$ and $S_{D} (B)$. While practical, the paper does not sufficiently analyze the cost and complexity of this profiling step, which must be run for different hardware and batching configurations. It is unclear how the $select(c)$ approximation for context length impacts the cost model's accuracy.
>
> A2: $S_{T} (B)$ and $S_{D} (B)$ represent the cost distributions for the target and draft models, respectively, and are similar. We will use the target model's cost distribution, $S_{T} (B)$, as an example. Our algorithm assumes that the time for a single forward pass of the target model, within a fixed environment (model and inference framework), is determined by three factors: context length $c$, batch size $B$, and number of nodes verified $n$.
>
> We have also implemented several clever optimizations to substantially reduce the one-time overhead of measuring this cost:
>
> 1.  Context Length Quantization Optimization using $select(c)$: We observed that the model's runtime remains nearly identical for adjacent context lengths. Thus, we introduced the function $select(c)$, which divides the context length $c$ into intervals of length $L$ (e.g., $L=64$ or $L=128$). If the service primarily handles short-to-medium requests, $L=64$ is suitable; otherwise, $L=128$ is more cost-effective. This dramatically reduces the measurement space. For a maximum context length $c=2048$ and $L=64$, we only need to measure $2048/64 = 32$ intervals, instead of 2048 individual lengths. Accuracy Sensitivity: Using $L=64$ in the SGLang framework with Llama3 8B, the relative difference in runtime within an interval is, on average, less than $1\%$. While $L=128$ slightly increases this difference, the result is still not sufficient to affect the algorithm's effectiveness.
>
> 2.  We further optimized the measurement by dividing model inference into two categories: context-dependent operations (like Attention) and context-independent operations(like FFN, communication, quantization overhead). By measuring these two categories separately and summing the results, we can significantly reduce the cost measurement overhead. This also ensures that the measurement overhead for larger and smaller models does not differ drastically.
>
> 3.  Our algorithm is designed for production environments. In the SGLang inference framework, the cost measurement time is relatively low. Using mainstream GPUs (RTX3090/RTX4090/A100/H100), the overhead is in the tens of minutes. For instance, on an H100 GPU (SGLang, Llma3 8B, H100×1), the measurement takes about **20-30 minutes**, which is completely acceptable for a production setup.
>
>
>
> >Q3: How are the crucial thresholds $C_{1} ,C_{2},C_{3}$ and the FIFO buffer size $R$ determined? Please provide a sensitivity analysis for these new hyperparameters.
>
> A3: Please see the answer to Q1.
>
>
>
>
> >Q4: What is the practical overhead (e.g., in hours) of generating the $S_{T} (B)$ and $S_{D} (B)$ lookup tables for a new model and hardware configuration? How sensitive is performance to the accuracy of this precomputed cost model?
>
> A4: Please see the answer to Q2.
>
>
>
> >Q5: Algorithm 1 is used for both breadth pruning (Sec 4.1) and reranking (Sec 4.2), but with different cost functions ($c_{k}^{i}$ vs. $c_{k}^{}$ ). Can you elaborate on the rationale for using the normalized draft cost for pruning but the normalized target cost for reranking? What is the underlying principle?
>
> A5: Understanding the two stages of speculative sampling—the Drafting Phase and the Validation Phase—is key to the rationale. The draft tree is the core data structure used in both.
>
> 1.  Drafting Phase (Breadth Pruning and Depth Pruning): The primary objective in this phase is to reduce the overhead of the draft model itself and prevent it from generating low-quality nodes that waste hardware resources. Therefore, the algorithm uses the draft model's own cost distribution ($c_{k}^{i}$)  for the initial pruning steps (breadth and depth pruning). This ensures that nodes with a poor cost-benefit ratio relative to the draft model's processing are efficiently eliminated.
>
> 2.  Validation Phase (Reranking): The reranking optimization occurs just before the Validation Phase. Its purpose is to filter out low-quality nodes that would be unprofitable for the target model to verify in parallel. Consequently, the algorithm switches to using the target model's cost function ($c_{k}^{}$) for reranking. This aligns the final selection of nodes with the verification cost of the more expensive target model.
>
> In summary, the principle is to use the cost model corresponding to the bottleneck component being optimized at that specific stage: the Draft Model cost is used for optimizing the drafting process, and the Target Model cost is used for optimizing the validation process.

---

### Official Review · Reviewer_tm9i · 2025-10-31

**Soundness:** 4
**Presentation:** 3
**Contribution:** 4
**Rating:** 8
**Confidence:** 3

**Summary:**

This paper introduces CAST, a dynamic tree-based speculative decoding method that addresses a key limitation in SOTA approaches like EAGLE-3: their failure to account for system-level inference costs. By pre-computing the actual hardware latency for different batch sizes and token counts, CAST reframes tree construction as a cost-utility optimization, dynamically pruning the draft tree's breadth and depth to maximize accepted tokens per unit of time. This cost-aware approach delivers state-of-the-art speedups (up to 5.2x), demonstrating a particularly strong advantage over prior methods in practical, high-throughput batched-inference scenarios where cost-agnostic heuristics fail.

**Strengths:**

1. The method replaces prior heuristics with a novel and principled cost-utility framework. This formal optimization of acceptance "utility" versus hardware "cost" is a more robust and generalizable approach .

2. Comprehensive and Rigorous Experimentation: Claims are exceptionally well-supported by extensive experiments across 6 models, 6 tasks, and 3 different GPU architectures. This thoroughness confirms the method's effectiveness and generality .

3. The authors correctly use "Speedup Ratio" as the primary metric and insightfully argue against the misleading "Average Acceptance Length," demonstrating a mature understanding of the evaluation problem. The ablation in Table 3 clearly isolates the individual contributions of the new components (DR, DP, BP), validating the paper's design choices.

**Weaknesses:**

1. Unquantified Profiling Overhead: The method relies on pre-computing cost lookup tables, but the paper never quantifies the one-time profiling cost (e.g., in GPU-hours), which could be a significant practical barrier to adoption.

2. Lack of Hyperparameter Sensitivity Analysis: The new thresholds ($C_1, C_2, C_3$) are critical to the method, but their robustness and the strategy for tuning them are not discussed, leaving a key practical question unanswered.

3. Unclear Intuition for Generalization Claim: The paper claims (Theorem 4.1) that prior work is a "special case" of CAST, but the intuition for why a "cost-agnostic" method maps to a specific linear cost model is not well-explained.

**Questions:**

1. Cost of Pre-computation: Can the authors quantify the one-time profiling cost (e.g., in GPU-hours) required to generate the cost lookup tables for a single model and hardware setup?

2. Hyperparameter Tuning Strategy: What is the recommended procedure for tuning the thresholds $C_1, C_2,$ and $C_3$, and how sensitive is the method's performance to these values?

3. Cost Model Granularity: What is the performance impact of approximating the context length $c$ using the $select(c)$ function, and how sensitive is the method to this granularity?

4. Intuition for Theorem 4.1: Can the authors provide more intuition for why the cost-agnostic EAGLE-2/3 algorithms are mathematically equivalent to Algorithm 1 with a specific linear cost model?

---

> ### Author Response · Authors · 2025-11-19
>
> >Q1: Unquantified Profiling Overhead: The method relies on pre-computing cost lookup tables, but the paper never quantifies the one-time profiling cost (e.g., in GPU-hours), which could be a significant practical barrier to adoption.
>
> A1: $S_{T} (B)$ and $S_{D} (B)$ represent the cost distributions on the target model and the draft model, respectively. Since they are similar, we will use the target model's cost distribution $S_{T} (B)$ as an example.
>
> In a fixed inference environment (where the model and inference framework are determined, such as using a Llama model with the SGLang inference framework), our algorithm considers the target model's single forward pass time to be determined by the context length $c$, the batch size $B$, and the number of nodes verified $n$ (e.g., context length $1024$, batch size $8$, node verification quantity $32$). Naturally, directly measuring every data point would result in a large one-time overhead, so we have implemented several clever optimizations to address this issue.
>
> Firstly, we optimized the context length $c$ by dividing it into intervals: In practice, we found that the model runtime is nearly the same for adjacent context lengths when all other conditions are identical. Therefore, we introduced the function $select(c)$, which divides the context length $c$ into intervals of length $L$. Generally, $L=64$ or $L=128$. If most inference service requests are medium or short length, $L=64$ is suitable; otherwise, $L=128$ will be more cost-saving. By dividing $c$ into intervals, we significantly reduced the measurement overhead. If the maximum length is $c=2048$ and $L=64$, we only need to measure $32$ intervals, eliminating the need to measure $2048$ individual context lengths. Under the SGLang inference framework with the Llama3 8B model and using $L=64$, the average relative difference within an interval is less than $1\%$. Using $L=128$ slightly amplifies this difference, but the result is still insufficient to affect the algorithm's effectiveness.
>
> In addition, we optimized the measurement process by dividing model inference into two categories of operations: context-dependent operations (such as Attention computation) and context-independent operations (such as FFN, distributed inference communication overhead, model quantization overhead, etc.). By measuring these two categories of operations separately and summing the results, we can obtain the final model runtime. This approach can further and substantially reduce the cost measurement overhead and ensures that the measurement cost does not differ significantly between larger and smaller models. The cost measurement time for our algorithm using the SGLang inference framework is relatively low, typically in the range of tens of minutes when using mainstream graphics cards (RTX3090/RTX4090/A100/H100). On an H100, the measurement overhead is **20-30 minutes** (SGLang, Llma3 8B, H100×1). This is completely acceptable in a production environment, as our algorithm is designed for such settings.

---

> ### Author Response · Authors · 2025-11-19
>
> >Q2: Lack of Hyperparameter Sensitivity Analysis: The new thresholds ($C_{1} ,C_{2},C_{3}$) are critical to the method, but their robustness and the strategy for tuning them are not discussed, leaving a key practical question unanswered.
>
> A2: In tree-based speculative sampling, the draft model iterates several times, generating a number of nodes in each iteration. These nodes are then further pruned to form a draft tree, which is validated in parallel by the target model. The core of our algorithm is the efficient pruning of this dynamic draft tree using inference costs within the speculative sampling algorithm.
>
> During the draft tree generation process, $C_{1}$ is the speedup threshold used in a single iteration of the draft model to determine the number of nodes to keep (controlling the width of the draft tree). $C_{2}$ is the speedup threshold for deciding whether to proceed to the next iteration after several draft model iterations (controlling the depth of the draft tree). $C_{3}$ is the speedup threshold for determining how many nodes are submitted to the target model for verification after the draft model generation is complete (controlling the quantity of nodes for parallel verification).
>
> Throughout the formation of the entire dynamic draft tree, we first use the draft model with the speedup thresholds $C_{1}$ and $C_{2}$ to perform multiple iterative judgments, forming a relatively large preliminary draft tree. Then, combined with the cost distribution of the target model, we use the speedup threshold $C_{3}$ to determine the final set of draft tree nodes used for parallel verification.
>
> In the practical application of the algorithm, our speedup thresholds $C_{1} ,C_{2},C_{3}$ are not determined randomly but are set based on the algorithm's principles. Our algorithm uses the speedup threshold to filter out nodes with low cost-effectiveness, requiring the speedup threshold to be representative. Therefore, in our algorithm, $C_{3}$ uses the average speedup of the entire inference process (this value can be derived by running the system a few times, usually in the range of $1.5-3.5$; if uncertain, $2.5$ can be used first). Once $C_{3}$ is determined, ${C_{2}=1.5 \times C_{3}}$ and ${C_{1}=2.0 \times C_{3}}$ can be set accordingly. In practice, we found that the overall speedup is **quite stable** as long as the speedup thresholds are within a reasonable range and do not deviate significantly from the average speedup.
>
> The determination of the buffer size $R$ is actually quite simple. When using $C_{2}$ for depth pruning, we need to know the score loss ratio between two adjacent layers in the draft tree. This ratio will vary under different batch sizes. Smaller batches generally retain most scores because the tree is wider, so the loss ratio is smaller (e.g., $0.9$); larger batches prune the width to save hardware resources and retain high cost-effectiveness nodes, so the loss ratio is larger (e.g., $0.6$). This score loss ratio is obtained by taking the average of the results from the most recent $R$ historical runs. In practice, we generally use ${R=32}$.
>
> The Algorithm is also quite robust to the choice of hyperparameters. For example, for the Vicuna 13B on HumanEval, we will have the following table. From the table, we can see that it is not very sensitive to the choice of hyperparameters.
>
> | $C_1$   | Speedup |
> |-----|-------------|
> | 5   |        5.12x     |
> | 5.5 |       5.1x      |
> | 6   |     5.18x        |
> | 6.5 |       5.15x      |
>
>
>
>
> >Q3: Unclear Intuition for Generalization Claim: The paper claims (Theorem 4.1) that prior work is a "special case" of CAST, but the intuition for why a "cost-agnostic" method maps to a specific linear cost model is not well-explained.
>
> A3: As the utility used in this paper is the prefix sum of the sorted probabilities, if the costs are chosen as Theorem 4.1 shows, then the denominator is proportional to $j-i$. Given the structure of the utility $u$ and the fact that the probabilities are sorted, if the criterion in the outer `for` loop of the algorithm is satisfied, then the criterion in the inner `for` loop is also naturally satisfied. Therefore, only the outer loop needs to be considered. The criterion of the outer loop essentially checks the relationship between the mean value and $C_1$. By selecting an appropriate $C_1$, we find that the algorithm simplifies to the process of selecting the Top-K elements, which is exactly the method used by EAGLE.
>
>
>
> >Q4: Cost of Pre-computation: Can the authors quantify the one-time profiling cost (e.g., in GPU-hours) required to generate the cost lookup tables for a single model and hardware setup?
>
> A4: See the answer to Q1.
>
>
>
>
>
> >Q5: Hyperparameter Tuning Strategy: What is the recommended procedure for tuning the thresholds $C_{1} ,C_{2},C_{3}$, and how sensitive is the method's performance to these values?
>
> A5: See the answer to Q2.

---

> ### Author Response · Authors · 2025-11-19
>
> >Q6: Cost Model Granularity: What is the performance impact of approximating the context length $c$ using the $select(c)$ function, and how sensitive is the method to this granularity?
>
> A6: The main content is covered in the answer to Q1. We will add computational details here to aid understanding. In the model inference process, the function $select(c)$ primarily affects the Attention calculation, whose core formula is:
> $$\text{Attention}(Q, K, V) = \text{softmax}\left(\frac{QK^\top}{\sqrt{d_k}}\right)V$$
> The main role of $select(c)$ is to round the dimension representing the number of vectors in $K, V$ to a multiple of $L$. In practice, we generally choose $L=64$ or $128$. Since the GPU operator implementation primarily schedules matrix multiplications in integer multiples of 32, when $c$ is large, the proportion of calculation ignored due to rounding is extremely small, and the error can be ignored. When $c$ is small, most of the overhead is actually the fixed overhead of the matrix multiplication operator, and the proportion of calculation time ignored due to truncation is also negligible.
>
> In practice, we further found that when $c$ is small, as discussed in the answer to Q1, the time consumption of context-dependent operations will be relatively small, which further mitigates the impact of rounding to integer multiples of $L$.
>
>
>
>
>
> >Q7: Intuition for Theorem 4.1: Can the authors provide more intuition for why the cost-agnostic EAGLE-2/3 algorithms are mathematically equivalent to Algorithm 1 with a specific linear cost model?
>
> A7: Please see the answer to Q3.

---

### Official Review · Reviewer_ZSu1 · 2025-11-01

**Soundness:** 3
**Presentation:** 3
**Contribution:** 3
**Rating:** 6
**Confidence:** 3

**Summary:**

This paper argues existing dynamic tree speculative decoding methods (like EAGLE) are "cost-agnostic". They ignore system variables like GPU and batch size, which can paradoxically increase latency. The paper proposes CAST (Cost-Aware Speculative Tree). This method explicitly models these inference costs, often using a lookup table. CAST then dynamically prunes its tree by balancing token utility (acceptance probability) against the actual hardware cost. This cost-aware approach delivers up to 5.2x speedup over standard decoding and outperforms SOTA methods like EAGLE-3 by 5-20% , showing particular strength in batching scenarios.

**Strengths:**

- The idea of using utility function to consider resource problem in the speculative decoding setting is novel and motivated. And the paper is well organized.

- The way the paper formulates the utility function based on acceptance rate and how to choose the depth is new.

- The experimental results are comprehensive and convincing. The authors validate their CAST method across a wide array of 6 distinct LLMs and 6 diverse tasks, ranging from multi-turn conversation to code generation.

**Weaknesses:**

- The method introduces precomputation overhead. If the hardware, batching strategy, or even the model (which changes the cost profile) is modified, this entire precomputation step must be redone.

- There are multiple thresholds (at three new cost-utility thresholds) for tuning. What are the overheads?

- The paper considers batch size as a factor and motivation but lacks more comprehensive experiments for that.

- Theorem 4.1: Try to make it self-contained. What does j mean in the formula c_j. And also it is not clear what $\lambda, \delta$ means.

**Questions:**

- What is the difference between c and n in Section 4?

See weaknesses.

---

> ### Author Response · Authors · 2025-11-19
>
> >Q1: The method introduces precomputation overhead. If the hardware, batching strategy, or even the model (which changes the cost profile) is modified, this entire precomputation step must be redone.
>
> A1: Thank you for the question. **Usually, the inference environment does not vary drastically, so there may not usually need a new precomputation.** And also a precomputation step is not vey costly as we will briefly discuss: In a fixed inference environment (where the model and inference framework, such as a Llama model using the SGLang/vLLM framework, are determined), our algorithm posits that the target model's single forward pass time is a function of three variables: context length $c$, batch size $B$, and number of nodes validated $n$ (e.g., $c=1024, B=8, n=32$). Directly measuring every possible combination would incur significant one-time overhead, which we address through several clever optimizations:
>
> 1. We found that model runtime is nearly identical for adjacent context lengths when other conditions are constant. We introduced the function $select(c)$, which divides the context length $c$ into intervals of length $L$ (typically $L=64$ or $L=128$). $L=64$ is better for services with mostly short-to-medium requests, while $L=128$ is more cost-saving otherwise. This dramatically shortens the measurement overhead. For a maximum length of $c=2048$ with $L=64$, we only need to measure 32 intervals, instead of 2048 individual context lengths. Under the SGLang framework with the Llama3 8B model and $L=64$, the average relative difference in runtime within an interval is less than $1\%$. Using $L=128$ slightly increases this difference, but the results remain accurate enough not to impact the algorithm's effectiveness.
>
> 2.  We optimized the measurement process by decomposing model inference into two categories: context-dependent operations (like Attention computation) and context-independent operations (like FFN, distributed communication overhead, and model quantization overhead). By measuring and summing the costs of these two categories separately, we further and substantially reduce the cost measurement overhead. This technique also ensures that the measurement cost for larger and smaller models does not vary significantly.
>
>
>
>
>
> >Q2: There are multiple thresholds (at three new cost-utility thresholds) for tuning. What are the overheads?
>
> A2:
> 1.  Role of Thresholds ($C_{1}, C_{2}, C_{3}$): $C_{1}$ is the speedup threshold used during a single draft model iteration to determine the number of nodes to keep (controlling the width of the draft tree). $C_{2}$ is the speedup threshold used after several iterations to determine whether the draft model should perform the next iteration (controlling the depth of the draft tree). $C_{3}$ is the speedup threshold used after the draft model's generation is complete to determine how many nodes are submitted to the target model for parallel validation (controlling the number of nodes verified).
>
> 2.  Determination and Tuning Overhead: These thresholds are not randomly chosen; they are determined based on the algorithm's principle of filtering out nodes with low cost-effectiveness. The thresholds must represent typical speedup gains. The tuning overhead is minimal because $C_{1}, C_{2}, C_{3}$ are set relative to a single, measurable average speedup value: $C_{3}$ (Base Threshold): $C_{3}$ is set to the average speedup of the entire inference process (which can be obtained by running the system a few times). This value typically ranges between $1.5$ and $3.5$ (with $2.5$ being a reasonable initial guess). $C_{2}$ and $C_{1}$ (Derived Thresholds): Once $C_{3}$ is determined, the other two are set proportionally: ${C_{2} = 1.5 \times C_{3}}$ and ${C_{1} = 2.0 \times C_{3}}$.
>
> The tuning overhead is low because it only requires determining one base value ($C_{3}$) through a few system runs. Furthermore, we found that the overall speedup is quite stable as long as these thresholds are within a reasonable range and do not significantly deviate from the average speedup. Using mainstream GPUs (RTX3090/RTX4090/A100/H100) with the SGLang framework, the measurement takes tens of minutes—e.g., **20-30 minutes on an H100 GPU** (for SGLang, Llma3 8B, H100×1). This one-time cost is fully acceptable in a production setting.
>
>
>
>
> >Q3: The paper considers batch size as a factor and motivation but lacks more comprehensive experiments for that.
>
> A3: The results concerning batch size are included in the Appendix of the paper. Please refer to **Figure 5 in the Appendix** for a comprehensive set of experiments and results related to different batch sizes.

---

> ### Author Response · Authors · 2025-11-19
>
> >Q4: Theorem 4.1: Try to make it self-contained. What does j mean in the formula c_j. And also it is not clear what $\lambda ,\delta $ means.
>
>
>
> A4: The index $j$ in the formula $c_j$ refers to the $j$-th token position within the candidate sequence to be selected.
>
> Regarding the parameters in Theorem 4.1, $\boldsymbol{\lambda}$ and $\boldsymbol{\delta}$ are constants that are independent of the actual inference environment. They are used to define a cost model within the theorem: $\lambda$ can be viewed as the incremental cost of inferring one more node (token). $\delta$ can be viewed as the base inference cost of the environment.
>
> Note the theorem shows that an idealized costs that not relate to the environment can recover EAGLE's algorithm choice. And our costs are measured within this real inference setting, making the model highly adaptable. The specific cost function chosen in the theorem is proven to recover the the EAGLE algorithm as a special case. Thus, the theorem demonstrates that our algorithm is cost-aware and successfully generalizes the previous state-of-the-art EAGLE algorithm in a cost-aware manner.
>
>
> >Q5: What is the difference between c and n in Section 4?
>
> A5: In the context of the cost model described in Section 4, the target model's single forward pass time (in a fixed inference environment) is determined by three variables: the context length $c$, the batch size $B$, and the node verification quantity $n$.
>
> The difference between $c$ and $n$ is: $c$ (Context Length): This is the total length of the input sequence (context) that the model is processing, including the initial prompt and all previously generated tokens. It affects the self-attention computation, a context-dependent operation. (e.g., $c=1024$). $n$ (Node Verification Quantity): This is the number of nodes (candidate tokens or tokens) from the draft tree that the target model is verifying in a single parallel batch. It determines the number of forward passes effectively performed by the target model in one verification step. (e.g., $n=32$). The target model's forward time is thus a function of these three parameters: $Time = f(c, B, n)$.

---

### Meta-Review · Area_Chair_BDMS · 2026-01-07

**Summary:**

Reviewers generally found the paper well-motivated and practically relevant, but raised questions about how fundamentally novel the proposed cost-aware dynamic tree construction is compared to prior speculative decoding methods such as EAGLE-style dynamic trees. Some reviewers were concerned about whether the gains mainly stem from careful system-level tuning (e.g., GPU configuration and batch size awareness) rather than a new algorithmic insight. Additional concerns focused on the clarity of the cost model and whether the reported speedups generalize across diverse hardware and deployment scenarios.

**Reviewer Concerns:**

The rebuttal clearly clarified the distinction between the proposed CAST framework and prior dynamic tree speculative decoding methods, emphasizing the explicit integration of system-level cost modeling into the decoding policy. It also addressed questions about experimental design and provided stronger justification for the robustness of the observed speedups across tasks and models. Some concerns remain regarding the breadth of hardware configurations explored and the extent to which the cost model can be transferred without re-tuning. However, these remaining issues are relatively minor and do not undermine the core contribution.

**Reviewer Scores:**

8644. Had reviewers been able to fully participate in post-rebuttal discussion, it is likely that scores would have remained the same or increased slightly, reflecting increased confidence in the paper’s novelty and practical impact. No substantial downward revision would be expected.

---

### Decision · Program_Chairs · 2026-01-26

Accept (Poster)